# Focal adhesion-derived liquid-liquid phase separations regulate mRNA translation

**Abhishek Kumar[1]\*[†], Keiichiro Tanaka[1], Martin A Schwartz[1,2,3]\***

[1]Yale Cardiovascular Research Center, Department of Internal Medicine (Division of Cardiovascular Medicine), New Haven, United States; [2]Department of Cell Biology, Yale School of Medicine, New Haven, United States; [3]Department of Biomedical Engineering, Yale University, New Haven, United States

## eLife assessment

In this **valuable** study, Kumar et al., provide evidence suggesting that the p130Cas drives the formation of condensates that sprout from focal adhesions to cytoplasm and suppress translation. Pending further substantiation, this study was found to be likely to provide previously unappreciated insights into the mechanisms linking focal adhesions to the regulation of protein synthesis and was thus considered to be of broad general interest. However, the evidence supporting the proposed model was **incomplete**; additional evidence is warranted to substantiate the relationship between p130Cas condensates and mRNA translation and establish corresponding functional consequences.

**\*For correspondence:**
akumar20@mdanderson.org
(AK);
martin.schwartz@yale.edu (MAS)

**Present address:** [†]Department of Thoracic Head & Neck Medical Oncology Research, The University of Texas MD Anderson Cancer Center, Houston, United States

**Competing interest:** The authors declare that no competing interests exist.

**Abstract** Liquid-liquid phase separation (LLPS) has emerged as a major organizing principle in cells. Recent work showed that multiple components of integrin-mediated focal adhesions, including p130Cas can form LLPS, which govern adhesion dynamics and related cell behaviors. In this study, we found that the focal adhesion protein p130Cas drives the formation of structures with the characteristics of LLPS that bud from focal adhesions into the cytoplasm. Condensing concentrated cytoplasm around p130Cas-coated beads allowed their isolation, which were enriched in a subset of focal adhesion proteins, mRNAs, and RNA binding proteins, including those implicated in inhibiting mRNA translation. Plating cells on very high concentrations of fibronectin to induce large focal adhesions inhibited message translation which required p130Cas and correlated with droplet formation. Photo-induction of p130Cas condensates using the Cry2 system also reduced translation. These results identify a novel regulatory mechanism in which high adhesion limits message translation via induction of p130Cas-dependent cytoplasmic LLPS. This mechanism may contribute to the quiescent state of very strongly adhesive myofibroblasts and senescent cells.

## Introduction

Integrin-mediated adhesions are dynamic structures comprising the integrins themselves plus a complex array of intracellular cytoskeletal linkers, adapters, and signaling proteins through which cell adhesion regulates a vast range of cellular functions, including cell survival and growth, differentiation, motility, and gene expression. Under conditions of high adhesion and contractility, integrins drive the the assembly of large focal adhesions that anchor contractile actomyosin stress fibers, whereas, under conditions of lower adhesion or extracellular matrix stiffness, cells form smaller more dynamic adhesions. Large adhesions generally promote contractility and cellular quiescence, while small dynamic adhesions are associated with cell motility and growth.

LLPS has emerged over the past decade as a major principle of intracellular organization (*Hyman et al., 2014*). LLPS is driven by highly multivalent, weak interactions between proteins or proteins and

RNAs (*Alberti et al., 2019*). These domains, also termed 'condensates,' exhibit highly dynamic behaviors, including rapid exchange of components between the droplets and the bulk phase, fusion, and splitting. LLPS can organize biochemical reactions and regulate a wide array of cell functions including gene expression, organelle structure and function, protein synthesis, and mechanotransduction to name a few.

Integrin-mediated adhesions share multiple features with LLPS. They can assemble, grow, split, fuse, and disassemble in live cells, which often correlate with cell functions such as migration and gene expression (*Burridge and Chrzanowska-Wodnicka, 1996*; *Geiger et al., 2009*; *Wehrle-Haller, 2012*). Adhesion components exchange rapidly with the cytoplasm (*Hoffmann et al., 2014*; *Parsons et al., 2010*; *Stutchbury et al., 2017*). These features prompted us to hypothesize that focal adhesions may contain phase-separated cytoplasmic proteins that mediate these behaviors. p130Cas (BCAR1) is an attractive candidate as it participates in multivalent interactions and contains extended disordered regions that often mediate condensate formation (*Case et al., 2019*; *Defilippi et al., 2006*; *Harte et al., 1996*; *Polte and Hanks, 1995*). We, therefore, began examining the behavior of GFP-tagged p130Cas in cells.

While these experiments were underway, *Case et al., 2022* reported that purified focal adhesion kinase (FAK) forms LLPS in vitro. They further reported that the addition of paxillin enhances FAK condensates; that phosphorylated 130cas, Nck, and N-Wasp also form condensates; and that FAK condensates and p130Cas-Nck-NWasp condensates synergistically recruit paxillin and kindlin-integrin complexes. Mutations in FAK and p130Cas that reduce LLPS also reduce adhesion formation in cells. Tensin1 (*Lee et al., 2023*), LIMD1 (*Wang et al., 2021*), and βPIX and GIT1 *Zhu et al., 2020* have also been reported to form condensates in vitro and in cells that localize to focal adhesions, the cytoplasm, and other compartments such as dendritic spines. Experiments using integrin cytoplasmic domains bound to supported lipid bilayers generated two-dimensional LLPS with focal adhesion proteins (*Hsu et al., 2023*). Mutations in LIMD1 and βPIX or GIT1 that interfere with domain formation perturbed focal adhesion functions including migration and durotaxis. These biochemical and cellular experiments thus established a strong connection between integrin-mediated adhesions and LLPS.

Our examination of p130Cas in cells unexpectedly revealed that this protein drove the formation of condensates that traffic in and out of focal adhesions, contain numerous proteins and RNAs involved in mRNA translation, and are distinct from known cytoplasmic structures. Functional analysis revealed that the formation of these condensates is regulated by cell adhesion and suppresses protein synthesis under conditions of high adhesion.

## Results

### p130Cas phase separation

Examination of the p130Cas amino acid sequence revealed two intrinsically disordered regions of the type often associated with LLPS (*Figure 1A*). The substrate domain, which contains 15 tyrosines potentially phosphorylated by src family kinases, forms the largest unstructured region, while a segment of the C-terminal domain is also disordered. These features are highly conserved in mouse p130Cas (*Figure 1—figure supplement 1A* and comments in the *Figure 1—figure supplement 1* legend). This finding prompted us to express p130Cas N-terminally tagged with EGFP and visualize its dynamic behavior in mouse embryo fibroblasts recently plated on fibronectin-coated coverslips (*Figure 1B*). Western blotting cell lysates for p130Cas and correcting for transfection efficiency showed that the transfected protein was ~4–5 x endogenous (*Figure 1—figure supplement 1B*). Live imaging revealed that, in addition to its localization to focal adhesions, small spots of p130Cas emerged from focal adhesions and moved into the cytoplasm and sometimes back to focal adhesions (*Videos 1–3* and *Figure 1—figure supplement 1C*). CHO, HeLa, and HEK293T cells showed similar behavior (*Figure 1—figure supplement 1D–F*). Staining for endogenous p130Cas did not show obvious cytoplasmic droplets in untransfected MEFs, however, MCF7 cells, which have ~2–2.5 x higher endogenous p130Cas than NIH3T3, showed cytoplasmic spots without overexpression (*Figure 1—figure supplement 1G and H*). Spots were observed merging to form a single circular droplet (*Figure 1C*), a behavior typical of LLPS. Droplet area increases over 24 hr after cell plating (*Figure 1D*, *Figure 1—figure supplement 1I*).

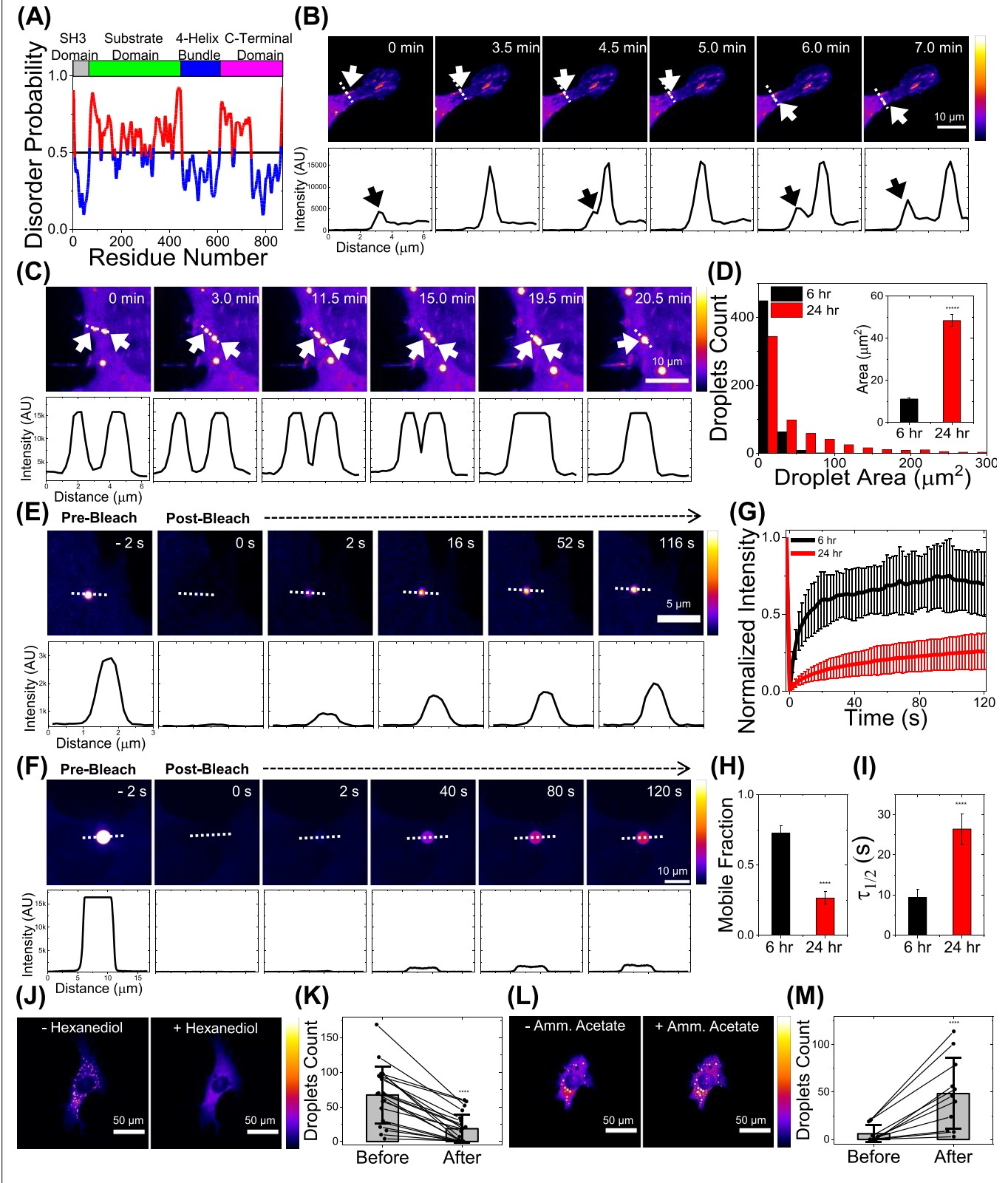

**Figure 1.** p130Cas phase separates in live cells. (**A**) Plot of p130Cas (*Homo sapiens*) amino-acid position versus predicted disorder probability using web-based tool Protein DisOrder (PrDOS). Red indicates high probability of disorder. Top: Color-coded protein domains. Horizontal line at 0.5 represents 5% false positive rate prediction. (**B**) Time-lapse images showing the emergence of a p130Cas droplet from a focal adhesion (FA) within a live NIH3T3 cell. Intensity scale bar to the right; black/blue represents low intensity while white/yellow represents high intensity. Corresponding intensity line

*Figure 1 continued on next page*

*Figure 1 continued*

profiles (drawn left (0 µm) to right) perpendicular to the FA long axis are shown in the lower panels. White arrow in image panel and black arrow in line profile indicate droplet formation and FA, respectively. Note that the p130Cas intensity in the droplet is significantly higher than in the cytoplasm or FAs. Scale bar = 10 µm. (**C**) Time-lapse imaging shows coalescence of two droplets in a live NIH3T3 cell; corresponding intensity line scan in the lower panel. Scale bar = 10 µm. (**D**) Histogram of p130Cas droplet area at ~6 hr (black) and ~24 hr (red) after plating transfected NIH3T3 fibroblasts on fibronectin-coated glass bottom dishes. N=521 droplets from 48 cells and N=631 droplets from 121 cells for 6 hr and 24 hr, respectively. Inset: Mean droplet area at 6 hr and 24 hr. Error bars = standard error of the mean (SEM). *P*-value <$5 \times 10^{-6}$ using Student's t-test. (**E–F**) Time lapse images showing droplet intensity (upper panel) and quantified intensity profile (lower panel) during FRAP at pre-bleach (–2 s), immediately after photobleach (0 s) and during recovery in cells at 6 hr (**E**) and 24 hr (**F**) after plating. Scale bar = 5 µm (**E**) & 10 µm (**F**). (**G–I**) Plot of normalized fluorescence intensity or recovery fraction with time for droplets at 6 hr (black curve) and 24 hr (red curve) and the corresponding mean mobile fraction (**H**) and $t_{1/2}$ for recovery (**I**) determined by fitting individual Fluorescence Recovery After Photobleaching (FRAP) curves to single component exponential recovery function. N=14 for 6 hr and N=7 for 24 hr. Error bars = standard deviation (SD) in (**G**) and SEM in (**H–I**). (**J–K**) Intensity coded image (**J**) and quantification of number of droplets (**K**) (N=21 cells) before and after treatment with 5% hexanediol for 2 min. Error bars are standard deviations. Scale bar = 50 µm. (**L–M**) Intensity coded image (**L**) and quantification of number of droplets (**M**) (N=11 cells) before and after treatment with 100 mM ammonium acetate for 8 min. Scale bar = 50 µm. Error bars = SD.

The online version of this article includes the following source data and figure supplement(s) for figure 1:

**Figure supplement 1.** p130Cas condensates in different cell lines.

**Figure supplement 1—source data 1.** Original membrane corresponding to *Figure 1—figure supplement 1B*, indicating the relevant bands.

**Figure supplement 1—source data 2.** Original membrane corresponding to *Figure 1—figure supplement 1G*, indicating the relevant bands.

To further characterize these cytoplasmic structures, we next measured the dynamics of EGFP-p130Cas in cytoplasmic droplets outside focal adhesions using Fluorescence Recovery After Photobleaching (FRAP). At 6 hr after plating, photobleached EGFP-p130Cas droplets recovered rapidly ($t_{1/2}$ = 9.4±1.9 s, mobile fraction ~72%; *Figure 1E, G and H*). FRAP at 24 after plating showed markedly slower recovery ($t_{1/2}$ = 26.4±3.7 s, mobile fraction ~26%; *Figure 1F–I*). Both the rapid molecular exchange at 6 hr and the decreased exchange over time support p130Cas LLPS in cells.

Phase separations can be modulated by reagents that alter the environmental polarity (*Alberti et al., 2019*; *Elbaum-Garfinkle, 2019*). We observed disruption of droplets after the addition of hexanediol (*Figure 1J&K*) and enhanced droplet assembly after the addition of ammonium acetate (*Figure 1L and M*; *Jain and Vale, 2017*), again, behaviors typical of phase separations. We conclude that p130Cas forms cytoplasmic condensates outside focal adhesions.

We next addressed the presence of other focal adhesion proteins in the p130Cas droplets. We stained EGFP-p130Cas-expressing cells for paxillin and FAK. Imaging at a focal plane through the nucleus and above the basal surface revealed strong co-localization of these proteins with p130Cas in cytoplasmic spots (*Figure 2A–D*). To assess dynamic behavior, we expressed EGFP-p130Cas with either tagRFP-paxillin or mcherry-FAK. These tagged constructs also co-localized in both focal adhesions and cytoplasmic droplets (*Figure 2—figure*

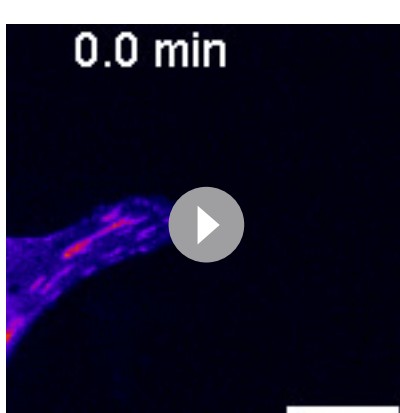

**Video 1.** Movie showing emergence of p130Cas droplets from focal adhesion (FA) in a live NIH3T3 fibroblast transiently transfected with EGFP-p130Cas freshly plated on fibronectin-coated glass bottom dishes. Time points are shown in each frame, frame interval = 30 s. Scale bar = 10 µm.

https://elifesciences.org/articles/96157/figures#video1

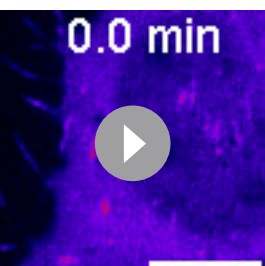

**Video 2.** Movie showing coalescence of two cytoplasmic p130Cas droplets in a live NIH3T3 fibroblast transiently transfected with EGFP-p130Cas freshly plated on fibronectin-coated glass bottom dishes. Time points are shown in each frame, frame interval = 30 s. Scale bar = 10 µm.

https://elifesciences.org/articles/96157/figures#video2

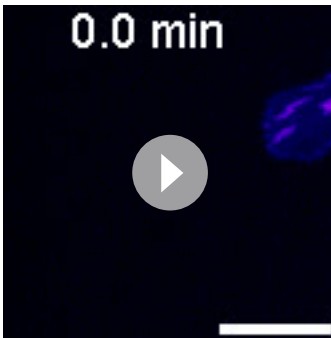

**Video 3.** Movie of p130Cas droplet merging back into a focal adhesion (FA) in a live NIH3T3 fibroblast transiently transfected with EGFP-p130Cas freshly plated on fibronectin-coated glass bottom dishes. Time points are shown in each frame, frame interval = 30 ss. Scale bar = 10 µm.

https://elifesciences.org/articles/96157/figures#video3

supplement 1A–C). Furthermore, the addition of ammonium acetate increased the colocalization of p130Cas with paxillin and FAK in cytoplasmic spots (*Figure 2—figure supplement 1D and E*). Paxillin (*Figure 2E*, *Video 4*) and FAK (*Figure 2F*, *Video 5*) co-emerged from focal adhesions along with p130Cas, indicating correlated dynamics. To test whether paxillin or FAK were required for the assembly of p130Cas droplets, we expressed EGFP-p130Cas in paxillin-null and in FAK-null MEFs. Cytoplasmic p130Cas droplets formed normally in both of these cell lines (*Figure 2G and H*), indicating that neither was required for p130Cas phase separation. By contrast, when tagged paxillin or FAK were expressed in p130Cas-null MEFs, cytoplasmic droplets were not evident (*Figure 2I and J*). These data argue that p130Cas is the main driver of phase separation outside focal adhesions.

We next addressed whether p130Cas within droplets was phosphorylated on its substrate domain tyrosines. Cells expressing EGFP-p130Cas stained for pY-Cas using a specific antibody (*Yaginuma et al., 2020*) revealed strong staining of condensates as well as focal adhesions (*Figure 2K–M*). We noticed, however, that pY-Cas staining relative to total p130Cas intensity was drastically higher in focal adhesions compared to droplets (*Figure 2N*). This result suggests that p130Cas is much less phosphorylated in droplets than in focal adhesions. We, therefore, considered whether phosphorylation of substrate domain tyrosines was required for phase separation. We examined mutants, including mutation of the 15 substrate domain tyrosines to phenylalanine, deletion of the substrate domain, and deletion of the disordered segment in the C-terminal domain (*Figure 2O*). Mutation of substrate domain tyrosines modestly decreased, deletion of the substrate domain strongly decreased, and deletion of the C-terminal disordered sequence had no effect on droplet formation (*Figure 2P and Q*). Thus, while the substrate domain is the main driver of phase separation, its tyrosine phosphorylation contributes to this effect to a lesser extent. The Y to F mutant substrate domain showed no change in intrinsic disorder (*Figure 2R*), consistent with roles for both phosphorylation-dependent and -independent interactions.

## Phase separation in vitro

We next considered whether p130Cas LLPS could be isolated and their composition determined. Due to the high protein exchange rates, the isolation of droplets from cells is impractical. We, therefore, attempted to induce phase separation in concentrated cell lysates prepared from suspended cells. We began by expressing EGFP or EGFP-p130Cas in HEK293T cells and isolating the transfected proteins on GFP-Trap beads. We then prepared lysate from cells expressing RFP-paxillin. Beads and lysates were mixed and left untreated, treated with ammonium acetate to induce phase separation, or with hexanediol to limit phase separation. Beads were then imaged without washing to avoid disassembly of condensed phases. We observed a low but detectable RFP-paxillin signal around EGFP-p130Cas beads under control conditions, a large increase after the addition of ammonium acetate and a decrease after the addition of hexanediol (*Figure 3A and B*). Paxillin was rapidly lost upon washing, indicating rapid dissociation. No paxillin fluorescence was observed around EGFP-only beads under any of these conditions. These results suggest a specific assembly of condensed phases on the bead-bound p130Cas. To capture bound components, 4% paraformaldehyde was added to the bead suspensions under these conditions, which were then washed (*Figure 3C and D*). Imaging showed paxillin intensity increased on EGFP-p130Cas beads after ammonium acetate treatment and no paxillin fluorescence was detected after hexanediol treatment, similar to the treatments in live

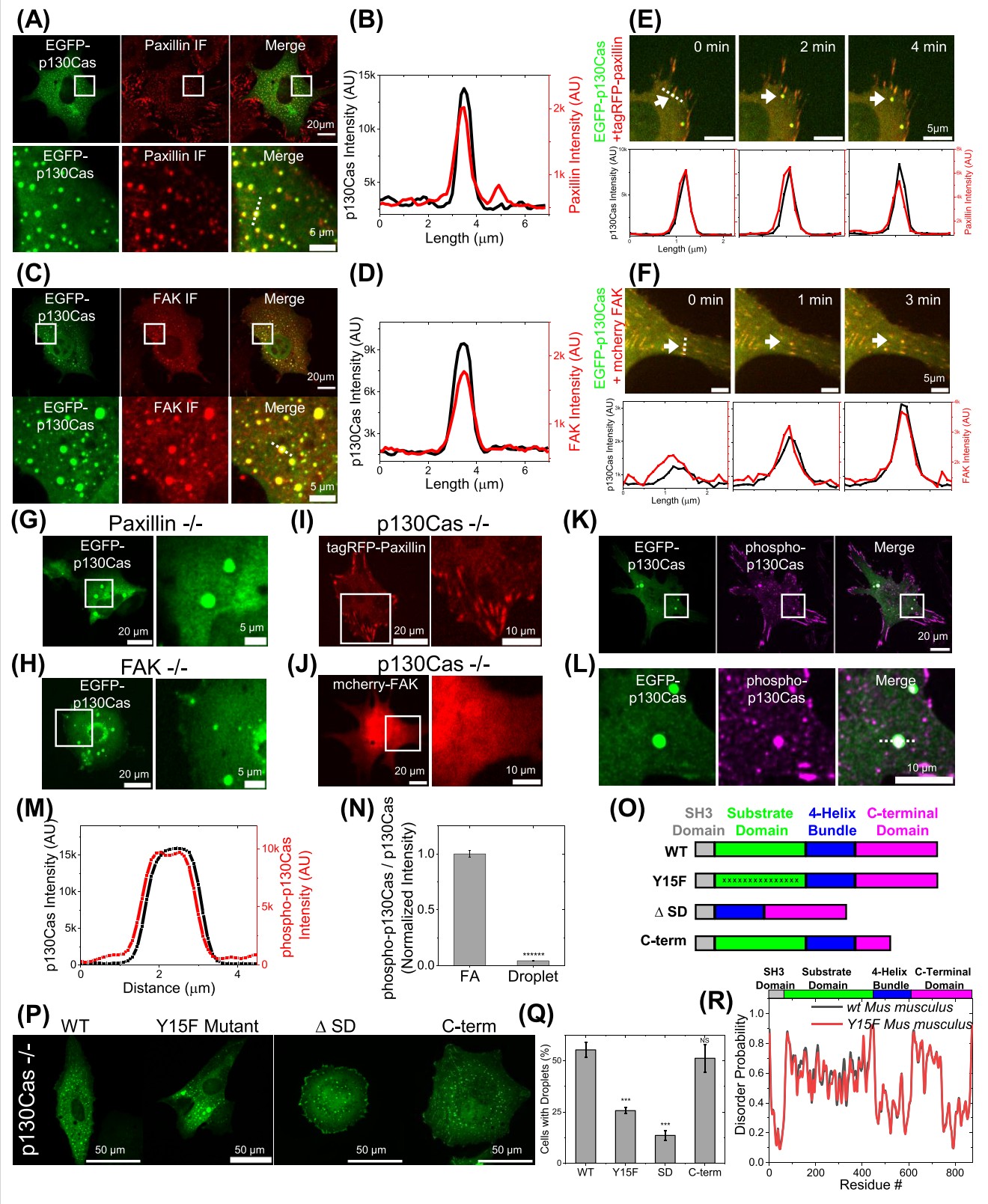

**Figure 2.** Paxillin and focal adhesion kinase (FAK) in p130Cas liquid-liquid phase separation (LLPS). (**A and C**) EGFP-p130Cas (green) transfected cells immunostained for paxillin (red) (**A**) and FAK (**C**). Lower panels show zoomed-in images of the area in the white box. (**B and D**) Line scans of p130Cas and paxillin (**B**) /FAK (**D**) showing co-localization. (**E and F**) Time lapse images of a cell co-transfected with EGFP-p130Cas (green) and either tagRFP-paxillin (red) (**E**) or mcherry-FAK (red) (**F**) showing the co-emergence from a focal adhesion (FA) of paxillin (**E**) and FAK (**F**) with p130Cas. Lower panel:

*Figure 2 continued on next page*

*Figure 2 continued*

corresponding line intensity profile. (**G and H**) EGFP-p130Cas in a paxillin-null (**G**) and FAK-null (**H**) cell. Right panels show zoomed-in images of the boxed area. (**I and J**) Image of a p130Cas-null cell transfected with either tagRFP-paxillin (**I**) or mcherry-FAK (**J**). Right panels show zoomed-in images of the boxed area. (**K**) Immunofluorescence image of a cell expressing EGFP-p130Cas (green) stained for phosphorylated p130Cas (purple). (**L**) Lower panels show zoomed-in images of the boxed area. (**M**) Corresponding intensity line profile of phosphorylated p130Cas and p130Cas. (**N**) Ratio of phosphorylated p130Cas to total p130Cas in FA and droplets. N=2176 FA and 53 droplets from 10 cells each. Error bars = SEM. (**O**) Schematic of WT, Y15F mutant, substrate domain deleted (Δ68–456 or ΔSD) and C-terminal domain partly deleted p130Cas (Δ611–742 or C-term). (**P**) Image of p130Cas-/- cells transfected with EGFP- WT-, Y15F, ΔSD/Δ68–456, or C-term/ Δ611–742 p130Cas. (**Q**) Quantification of percentage of droplet-positive cells for the indicated constructs from N=7 (N indicates independent experiments) with 146/238, 76/156, 231/347, 56/86, 205/396, 236/430, and 110/284 cells; N=4 with 82/278, 52/201, 42/184, and 58/238 cells; N=3 with 38/346, 23/203, and 68/371 cells and N=3 with 139/287, 142/222, and 134/325 cells, respectively. Error bars = SEM. (**R**) Plot of disorder probability versus amino acid for WT (black) and Y15F mutant (red) mouse p130Cas. Top: Color-coded protein domains.

The online version of this article includes the following figure supplement(s) for figure 2:

**Figure supplement 1.** Colocalization of p130Cas condensates with paxillin and FAK.

cells. This protocol thus allows the capture of cytoplasmic components that specifically associated with p130Cas condensates.

We next performed this procedure using the reversible crosslinker dithiobis succinimidyl propionate (DSP). Beads were washed, crosslinking reversed by treating with β-mercaptoethanol, and samples were analyzed by mass spectrometric proteomics. Comparing EGFP-p130Cas to EGFP-only as a control, peptides with at least two counts and twofold enrichment relative to control beads were considered specific (*Supplementary file 1*). Out of the 78 proteins reported to interact directly with p130Cas (*Evans et al., 2017*), 30 were enriched in the condensates (*Figure 3—figure supplement 1*). For the proteins specifically isolated on EGFP-p130Cas beads, the most prominent Gene Ontology (GO) terms for biological processes identified protein synthesis, RNA splicing, and RNA processing (*Figure 3E*).

## p130Cas condensates and RNA interactors

The connection to RNA metabolism led us to consider the relationship of p130Cas LLPS to stress granules and p-bodies that are well-known to regulate mRNA functions (*Freibaum et al., 2021*). p130Cas showed little correlation with the stress granule marker G3BP2, either without or with arsenite treatment to induce their assembly (*Figure 4A–D*). We also saw that of the six major stress granule constituents CAPRIN1, PRRC2C, USP10, UBAP2L, and CSDE1 (*Yang et al., 2020*) and PABPC1 (*Kedersha et al., 2000*) only PABPC1

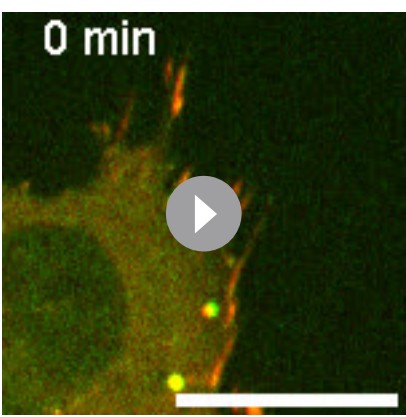

**Video 4.** Movie of p130Cas droplet co-emerging from focal adhesion (FA) along with paxillin. EGFP-p130Cas is in green and tagRFP-Paxillin is in red. Time points are shown in each frame, frame interval = 2 min. Scale bar = 10 μm.

https://elifesciences.org/articles/96157/figures#video4

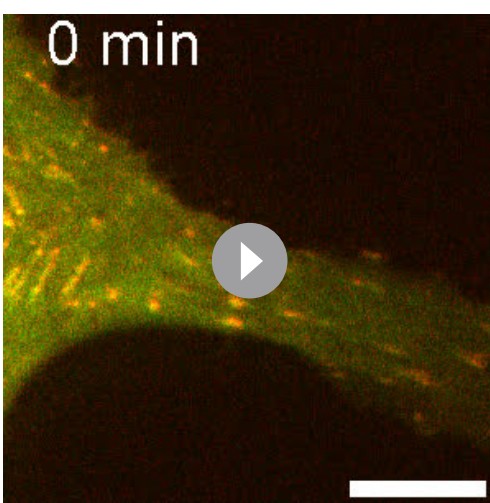

**Video 5.** Movie of p130Cas droplet co-emerging from focal adhesion (FA) along with focal adhesion kinase (FAK). EGFP-p130Cas is in green and mcherry-FAK is in red. Time points are shown in each frame, frame interval = 1 min. Scale bar = 10 μm.

https://elifesciences.org/articles/96157/figures#video5

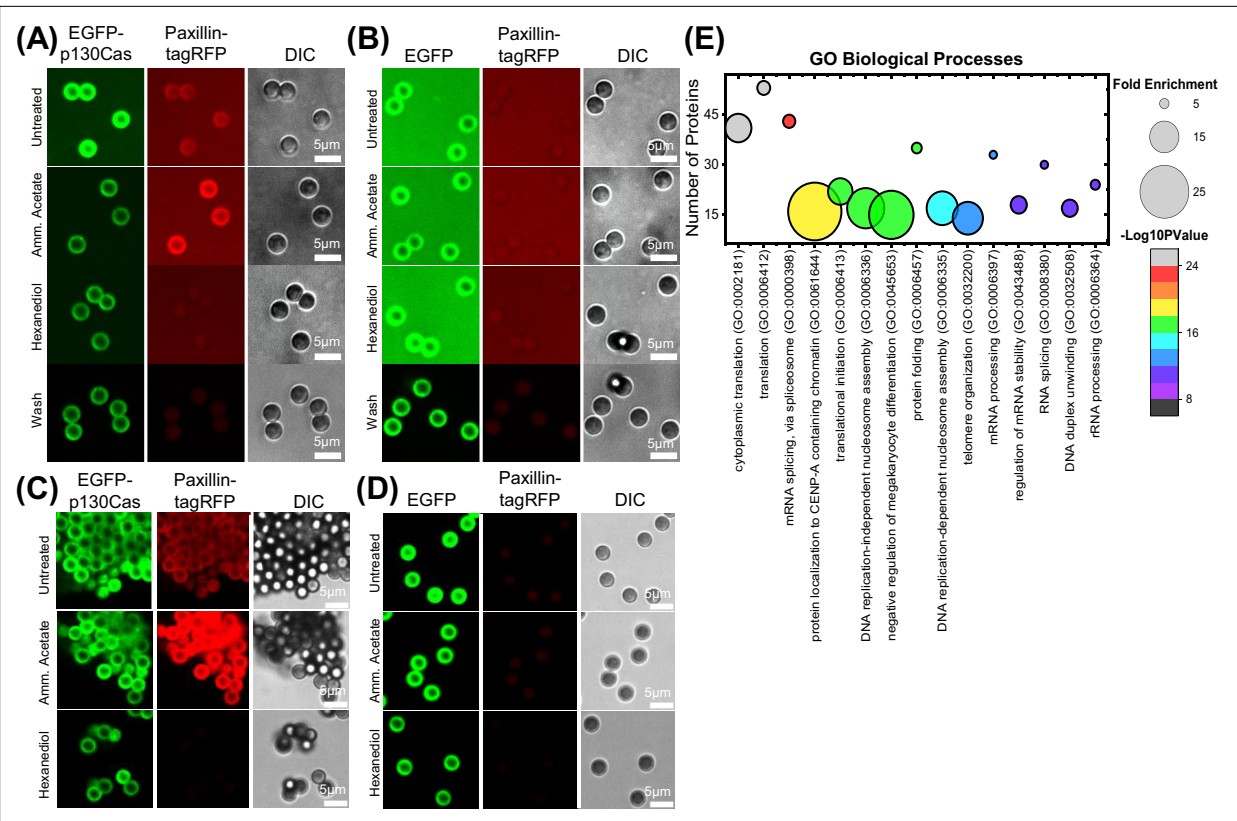

**Figure 3.** Isolation and characterization of p130Cas droplets. (**A and B**) Fluorescence image of GFP-trap magnetic beads incubated with cell lysate from HEK293tx cells transfected with either EGFP-p130Cas (**A**) or EGFP (**B**) and then mixed with lysate from tagRFP-paxillin transfected cells. Images show p130Cas on beads (green- left panel), paxillin on beads (red- middle panel) and DIC images of beads (gray- right panel): First row- Untreated, second row- 250 mM ammonium acetate, third row- 5% hexanediol, fourth row- washed with PBS. (C & D) GFP-trap beads incubated with cell lysate from HEK293tx cells transfected with either EGFP-p130Cas (**C**) or EGFP (negative control) (**D**) plus lysate from tagRFP-paxillin cells. Beads treated as in A & B were fixed with 4% paraformaldehyde and then washed with cold PBS. Note retention of paxillin after washing. (**E**) Proteins specifically associated with p130Cas beads (≥2 counts and twofold enrichment compared with control beads) were analyzed for Gene Ontology (GO; biological processes) using the online webtool-Database for Annotation, Visualization, and Integrated Discovery (DAVID). Number of proteins versus corresponding Gene ontology (GO) term is plotted with fold enrichment depicted by the size of the circle and -log10(p-value) by the color of the circle. The corresponding fold enrichment and -log10(p-value) scale bars are shown to the right of each plot. GO terms are sorted in descending order of their -log10(p-values).

The online version of this article includes the following figure supplement(s) for figure 3:

**Figure supplement 1.** Venn diagram of proteins in p130Cas condensates and its direct interactors obtained using mass spectrometric proteomics.

was detected in p130Cas condensates (*Supplementary file 1*). Poly(A)-binding protein (PABP), which can be present in stress granules, did not colocalize with p130Cas droplets immediately after induction of stress granules (*Figure 4E and F*). However, argonaute 2 (Ago2), which is critical for miRNA-dependent mRNA regulation (*Ender and Meister, 2010*) and is a constituent of stress granules (and p-bodies), partially overlapped with p130Cas-positive LLPS after arsenite treatment (*Figure 4G and H*). Dcp1A, a critical component of p-bodies, showed no overlap with p130Cas droplets (*Figure 4I–L* & *Figure 4—figure supplement 1A and B*). We conclude that p130Cas condensates share some constituents but are distinct from both stress granules and p-bodies.

We next explored the localization of RNA binding and processing proteins in p130Cas droplets. Ago2 is concentrated in the center of the droplets (*Figure 5A and B*). GW182, which, like Ago2, is a component of the RISC complex that mediates miRNA-directed mRNA cleavage (*Ding and Han, 2007*), was present in the outer rim of p130Cas structures (*Figure 5C and D*). Staining of the LLPS around EGFP-p130Cas beads in cell lysates confirmed the presence of Ago2 and GW182 (*Figure 5—figure supplement 1A and C* with *Figure 5—figure supplement 1B and D* as corresponding negative controls). Interestingly, Ago2 and GW182 appear to occupy subdomains within the p130Cas-positive zones around the beads, mimicking their localizations in cells. Messenger RNAs, which were localized

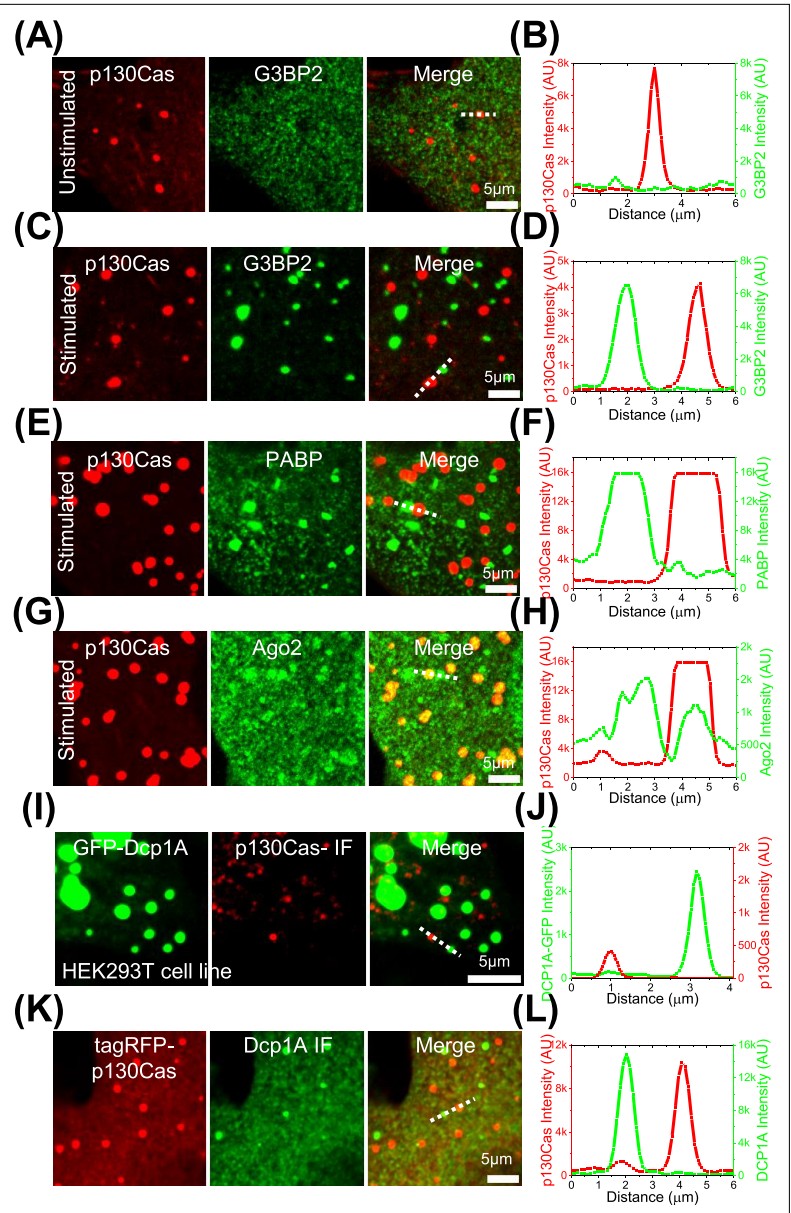

**Figure 4.** p130Cas comparison to stress granules and p-bodies. (**A, C, E and G**) Cells expressing EGFP-p130Cas (red- left panel) stained for G3BP2 (**A and C**), PABP (**E**), or Ago2 (**G**) (green - middle panel) and merged (right panel) under control/unstimulated (**A**) and stress granule induction with sodium arsenite (**C, E, and G**). (**B, D, F, and H**) The corresponding line intensity profile across a p130Cas droplet and a stress granule. (**I**) Immunofluorescence image of p130Cas (red - middle panel) in HEK293T cells stably expressing EGFP-Dcp1A marking p-granules (green-left panel) and the merge (right panel). (**J**) The corresponding line intensity profile across a p130Cas droplet and a p-granule. (**K**) Immunofluorescence image of Dcp1A (green - middle panel) in NIH3T3 cells expressing tagRFP-p130Cas (red-left panel) and the merge (right panel). (**L**) The corresponding line intensity profile across a p130Cas droplet and a p-granule.

The online version of this article includes the following figure supplement(s) for figure 4:

**Figure supplement 1.** Colocalization between p130Cas condensates and p-bodies.

by fluorescence in situ hybridization (FISH) using poly-T probes, yielded strong positive signals within a subset of p130Cas droplets in cells (*Figure 5E and F*). To test the relationship between protein synthesis and p130Cas droplets, cells expressing GFP-p130Cas were treated with cycloheximide (CHX), which blocks translational elongation and arrests mRNAs on the ribosomes, thereby depleting mRNAs from other pools (*Ivanov et al., 2019*; *Kedersha et al., 2000*; *Lui et al., 2014*; *Riggs et al.,*

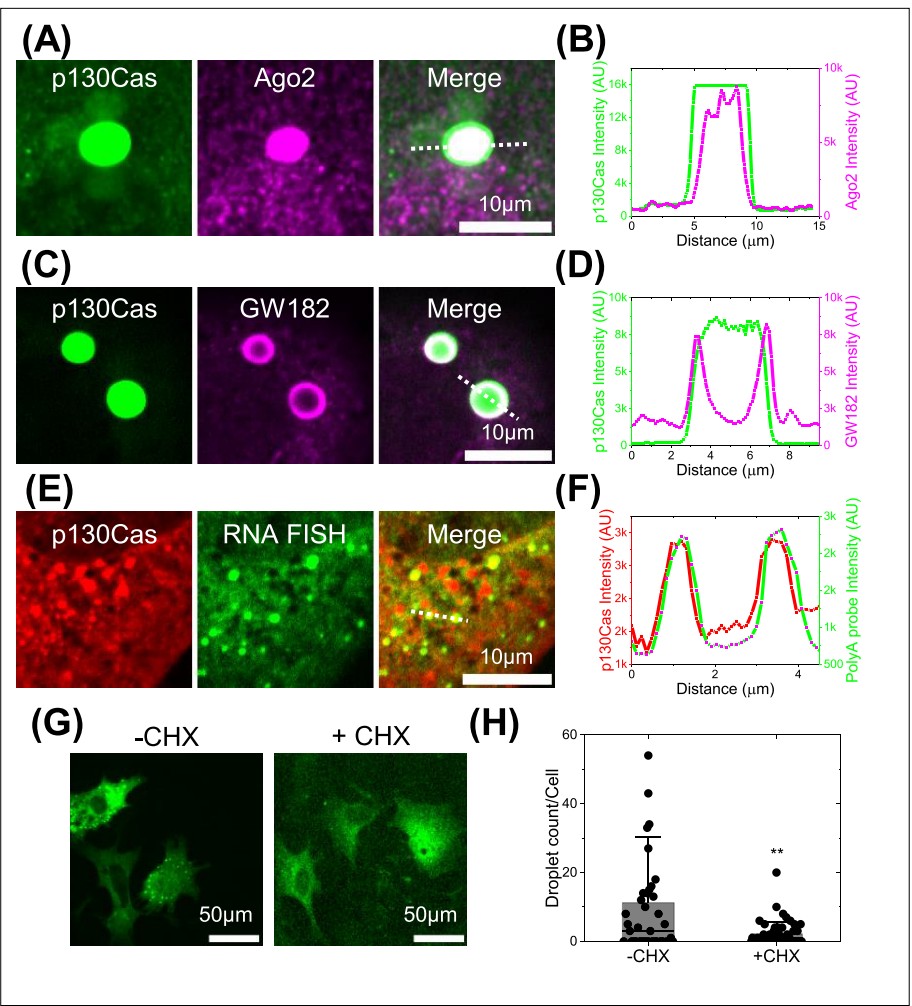

**Figure 5.** p130Cas comparison to mRNA processing proteins and mRNA. (**A and C**) Cells expressing EGFP-p130Cas (green- left panel) stained for Ago2 (**A**) and GW182 (**C**) (purple- middle panel) and merged (right panel). (**B and D**) Corresponding line intensity profile across a p130Cas droplet showing its colocalization with Ago2 present throughout the droplet (**A**) and GW182 at the periphery of the droplet (**C**). (**E**) Cells expressing tagRFP-p130Cas (red- left panel) with RNA- fluorescence in-situ hybridization (FISH) of poly-A binding probes (green- middle panel) and merged. (**F**) The corresponding line intensity profile across a p130Cas droplet showing colocalization with RNA-FISH probes. (**G**) EGFP-p130Cas expressing cells plated for 5 hr without (-CHX) or with (+CHX) 100 μg/ml cycloheximide for 2 min. (**H**) Number of droplets per cell under these conditions. N=3 independent experiments with 39 and 52 cells for -CHX and +CHX, respectively. Bar represents Mean, horizontal line is median and Error bars = SD.

The online version of this article includes the following figure supplement(s) for figure 5:

**Figure supplement 1.** Ago2 and GW182 in *in vitro* purified p130Cas condensates on beads.

*2020*). CHX decreased p130Cas LLPS (*Figure 5G&H*). Together, these results in *Figure 5* suggest that p130Cas condensates may regulate translation or other aspects of mRNA metabolism.

## Isolation of mRNAs in p130Cas condensates

Based on these results, the DSP-fixed in vitro p130Cas condensates vs the GFP-only control was reversed using β-mercaptoethanol, mRNAs isolated and analyzed by high throughput sequencing (RNAseq). RNAs that were enriched by > twofold with p-values <0.05 were considered positives (*Figure 6A* and *Supplementary file 2*). The most prominent Gene Ontology categories for differentially isolated genes identified functions related to cell cycle progression, survival, and cell-cell communication (*Figure 6B*). To test the functional relevance of these results, we next expressed

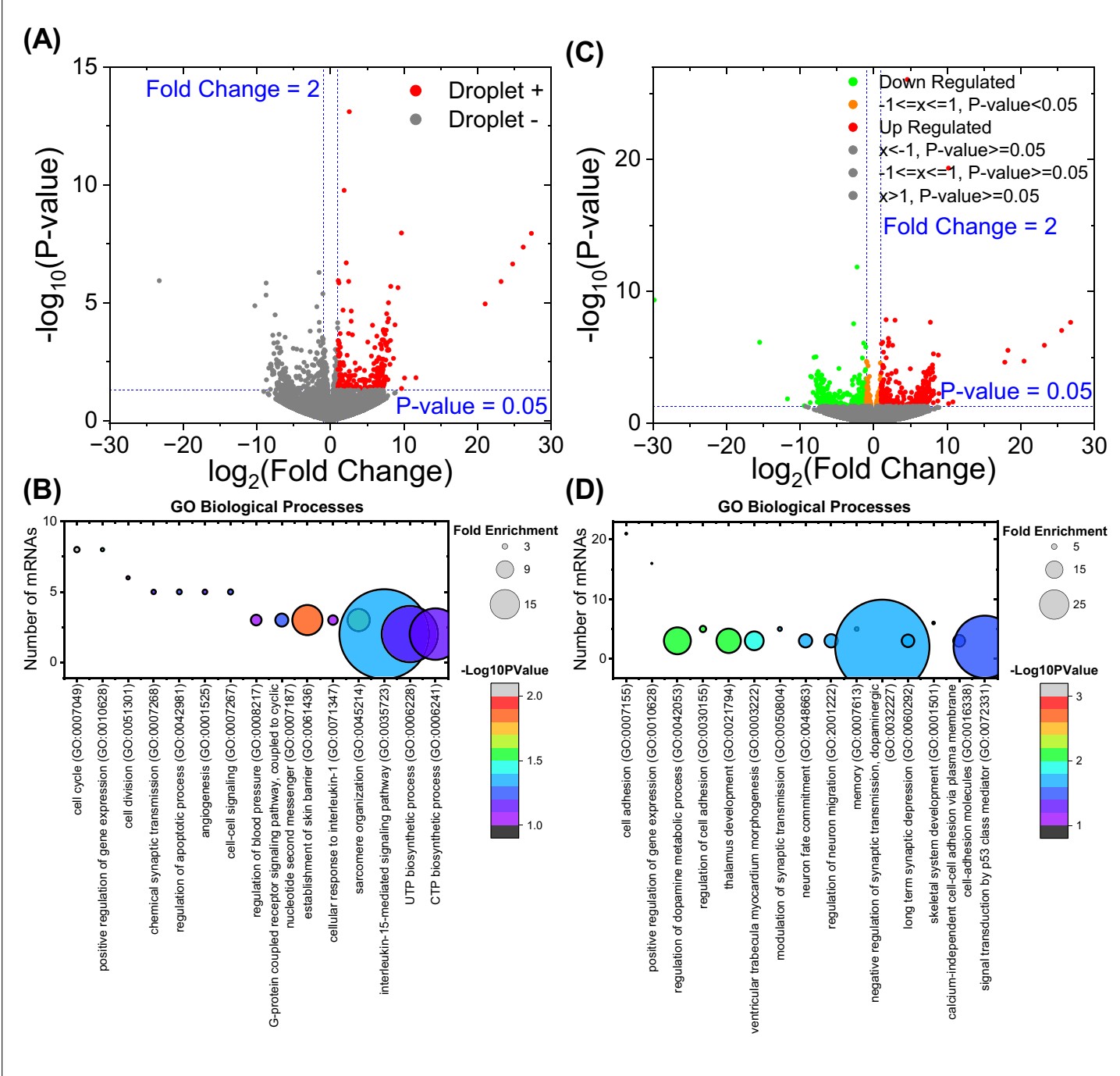

**Figure 6.** mRNAs associated with p130Cas droplets. (**A**) Volcano plot of mRNAs enriched on p130Cas beads compared to the EGFP-only control. mRNAs with twofold enrichment and p-value <0.05 are associated with p130Cas droplet (red dots). (**B**) Gene Ontology (GO) analysis (biological processes) of p130Cas-enriched mRNAs as described in Methods. (**C**) Volcano plot of p130Cas-regulated mRNAs. mRNAs with twofold up-regulation (red dots) or down-regulation (green dots) after p130Cas expression compared to the control transfected. (**D**) GO analysis (biological processes) of p130Cas-regulated mRNAs as described in Methods. Number of mRNAs versus corresponding GO term is plotted with fold enrichment depicted by the size of the circle and -log10(p-value) by the color of the circle. The corresponding fold enrichment and -log10(p-value) scale bars are shown to the right of each plot. GO terms are sorted in the descending order of the mRNA count (**B**) or -log10(p-values) (**D**).

GFP-p130Cas in the same HEK293T cells and carried out RNAseq to identify differentially expressed genes (*Figure 6C* and *Supplementary file 3*). Out of 297 mRNAs present in p130Cas droplets, 46 were differentially regulated after overexpression of EGFP-p130Cas. The probability of choosing these mRNAs at random is ~<5% (described in Methods), indicating a significant correlation between

droplet-associated and differentially regulated mRNAs. Cell adhesion and inflammatory gene expression were the most strongly affected pathways (*Figure 6D*).

## p130Cas LLPS modulates protein synthesis

Given these findings, we next considered whether cytoplasmic p130Cas phase separation might affect mRNA translation. With this in mind, we examined the breast cancer cell line MCF7, which, like many cancer cell lines, expresses higher levels of p130Cas than mouse embryo fibroblasts. We plated MCF7 cells on coverslips coated with low, medium, or high fibronectin (FN). To identify cytoplasmic phase separations, we stained cells for paxillin, due to the higher quality of antibodies to this protein compared to p130Cas. Again, focusing on a plane above the basal surface, we observed more cytoplasmic droplets on higher FN, as well as the expected increase in focal adhesions (*Figure 7A–C*). Next, we measured rates of protein synthesis using the puromycin assay (*Forester et al., 2018*; *Liu et al., 2012*). Puromycin interrupts translation by adding onto the growing peptide chain, thus, its incorporation is proportional to the number of mRNAs undergoing translation. Staining cells for puromycin and quantifying integrated signal per cell showed that translation decreased as a function of FN coating density (*Figure 7D and E*, *Figure 7—figure supplement 1A*) cycloheximide treatment as a negative control for puromycin labeling. We then asked if this effect required p130Cas. MCF7 cells were transfected with siRNA against p130Cas (knockdown confirmed in *Figure 7G*, *Figure 7—figure supplement 1B*) plated on low, medium, and high FN and subject to the puromycin assay (*Figure 7F–H*). Knockdown of p130Cas completely abrogated inhibition of translation on high FN, indeed, after p130Cas knockdown, high FN modestly increased translation. To test if this behavior is specific to this cancer line, we examined human umbilical vein endothelial cells (HUVECs) that express moderately high levels of p130Cas (*Figure 7I-K*, *Figure 7—figure supplement 1C and D*). In these cells, translation also decreased on high FN, which was again converted to an increase after p130Cas depletion. By contrast, NIH3T3 cells that have lower levels of p130Cas (*Figure 7—figure supplement 1E–G*) showed increased translation on high FN, with no effect of p130Cas knockdown. The differentiated breast epithelial line MCF10A, where p130Cas expression is low, also showed little change in puromycin labeling on low vs high FN (*Figure 7L and M*). Together, these results show that high p130Cas expression is required for a reduction in protein synthesis on high FN, an effect that correlates with droplet formation.

While these experiments implicate p130Cas in regulating translation, they do not address whether cytoplasmic phase separation is required. To address this issue, we expressed p130Cas fused to the light-inducible dimerizer Cry2, which is commonly used to drive phase separation by increasing the valency of protein interactions (*Courchaine et al., 2021*; *Shin et al., 2017*). In MCF7 cells expressing cry2-tagRFP-p130Cas, blue light-induced the formation of droplets that co-localized with paxillin and FAK, similar to native droplets (*Figure 8A and B*, *Figure 8—figure supplement 1A and B*). MCF7 cells expressing this construct (*Figure 8—figure supplement 1C and D*) and kept in the dark behaved similarly to control cells, with translation decreasing on high FN (*Figure 8C and D*). Irradiation with 488 nm laser light for 0.5 s induced rapid droplet formation, which in the absence of light then disassembled (t½ = ~9 min) (*Figure 8E and F* and *Video 6*). Next, these cells were left in the dark or irradiated for 2 hr using a pulsed light protocol (*Figure 8—figure supplement 1E*). Some cells were then incubated in the dark for the indicated times (*Figure 8G*). LLPS formed immediately after illumination (0 min) followed by disassembly over time in the dark (*Figure 8H*). Puromycin labeling revealed ~35% dip in protein synthesis after illumination which in the dark gradually returned to initial levels (*Figure 8I*). Cells were also subject to longer illumination using intermittent pulses (*Figure 8—figure supplement 1E*) to limit radiation damage (*Figure 8—figure supplement 1F and G*). Under this protocol, droplets were induced and puromycin labeling suppressed for the duration of the treatment (*Figure 8J–L*). These experiments provide direct evidence that p130Cas droplet formation inhibits translation.

## Discussion

Src family kinases, FAK or its homolog Pyk2, and/or paxillin have been localized to perinuclear or nuclear spots that traffic to and from focal adhesions in multiple systems (*Aoto et al., 2002*; *Day et al., 2021*; *Fincham et al., 1996*; *Seko et al., 1999*), though their characteristics and mechanisms were not further investigated. Recent work provides strong evidence for LLPS behaviors of the focal

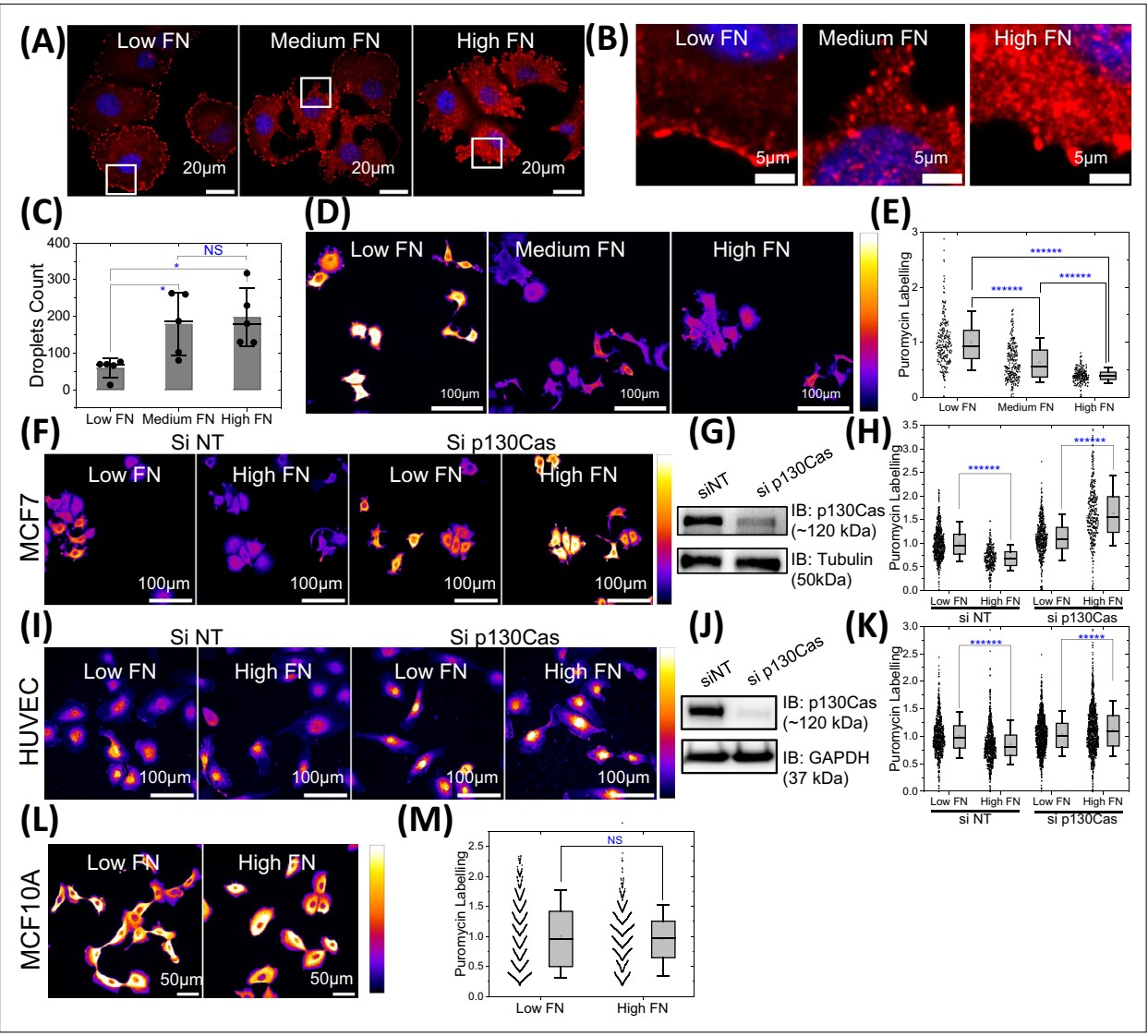

**Figure 7.** p130Cas liquid-liquid phase separation (LLPS) regulates global protein synthesis in hyper-adhesive cells. (**A–B**) MCF7 on low (2 µg/ml), medium (20 µg/ml), and high (50 µg/ml) fibronectin-coated dishes stained for paxillin (red) with the nucleus labeled using Hoechst-33343. Zoomed-in images (**B**) of the area labeled in the white box in (**A**) showing punctate paxillin in the cytosol to mark droplets. Scale bar = 20 µm (**A**) and 5 µm (**B**). (**C**) Quantification of puncta in MCF7 cells from A&B. N=5 field of views (FOVs) with 3–6 cells per field of view from three independent experiments imaged using high resolution 60x objective. (**D**) Intensity-coded images of MCF7 cells on low, medium, and high fibronectin (FN) incubated with puromycin then fixed and stained with anti-puromycin antibody. Scale bar = 100 µm. All images in a panel are shown at same intensity scale as depicted in the right color bar with black/blue as low intensity and white/yellow as high intensity. (**E**) Quantification of puromycin labeling intensity from D. N=262, 285, and 240 cells from 16 field of views each for low, medium, and high fibronectin, respectively from three independent experiments. (**F and I**) Intensity-coded images of puromycin labeling in MCF7 cells (**F**) & HUVECs (**I**) transfected with scrambled siRNAs or p130Cas siRNA on low and high fibronectin coated substrate. Scale bar = 100 µm. (**G and J**) Immunoblot of p130Cas for cells in (**G**) and HUVECs (**J**), with tubulin (**G**) and GAPDH (**J**) as loading controls. (**H and K**) Quantification of puromycin intensity for multiple cells MCF7 from F & I. N=690, 331, 547, & 359 MCF7 cells and N=792, 873, 1232, & 1054 HUVECs from 25 FOVs each from three independent experiments for cells on low and high fibronectin without and with p130Cas knockdown respectively. (**L**) Intensity coded images of puromycin labeling in MCF10A cells plated on low and high fibronectin coated substrate. Scale bar = 50 µm. (**M**) Quantification of puromycin intensity from multiple cells from L. N=859 and 863 cells from 25 field of views from three independent experiments for cells on low and high fibronectin, respectively.

The online version of this article includes the following source data and figure supplement(s) for figure 7:

**Source data 1.** Original membrane corresponding to *Figure 7G* (Same as *Figure 1—figure supplement 1—source data 2*), indicating the relevant bands and treatment.

**Source data 2.** Original membrane corresponding to *Figure 7J*, indicating the relevant bands and treatment.

**Figure supplement 1.** Role of p130Cas LLPS in modulating protein synthesis in hyperadhesive cells.

*Figure 7 continued on next page*

*Figure 7 continued*

**Figure supplement 1—source data 1.** Original membrane corresponding to *Figure 7—figure supplement 1F*, indicating the relevant bands and treatment.

adhesion proteins FAK and p130Cas (*Case et al., 2022*). Purified FAK undergoes LLPS in vitro, mediated mainly by its disordered N-terminus. They also observed LLPS of purified p130Cas phosphorylated on its substrate domain tyrosines via SH2 and SH3 domain interactions with Nck and N-WASP. Additional studies showed LLPS of focal adhesion components with evidence for regulation of focal adhesion dynamics and function (*Hsu et al., 2023*; *Lee et al., 2023*; *Wang et al., 2021*; *Zhu et al., 2020*).

Our current results show that focal adhesions also give rise to p130Cas-FAK-paxillin LLPS in the cytoplasm. These structures show typical LLPS behaviors including fusion and fission, and rapid exchange of subunits with the cytoplasm that slows as condensates mature. Phase separation was also modulated by reagents that alter polarity and affect LLPS in other systems. Cytoplasmic condensates contain little or no talin or vinculin but are enriched in RNA-binding proteins and mRNAs. However, p130Cas LLPS appears distinct from stress granules and P granules that contain RNAs and RNA-binding proteins. Formation of condensates was strongly reduced by deletion of the p130Cas substrate domain, with a more modest reduction by mutation of the 15 substrate domain tyrosines, indicating that, within cells, p130Cas LLPS involves both phosphotyrosine-dependent and -independent interactions.

LLPS is difficult to isolate due to the fast dissociation of their constituents. We, therefore, developed a method to induce condensation around magnetic beads coated with GFP-p130Cas vs. GFP alone, which could then be reversibly fixed, isolated, and components analyzed. While this method is no doubt imperfect, we saw large differences between p130Cas and control beads, and several components seen in vitro were confirmed in cells. We also noted that in some p130Cas droplets, different proteins localized preferentially to the outer vs inner regions. This behavior is common among LLPS and likely reflects functional domains, though further work will be required to understand these aspects.

The presence of RNA binding proteins and mRNAs in p130Cas condensates led us to investigate a possible link to mRNA translation. We found that in MCF7 cells and HUVECs, which express abundant p130Cas, adhesion to high levels of FN reduced total translation as measured by the puromycin incorporation assay. However, the knockdown of p130Cas converted this reduction into an increase. These results indicate that p130Cas is required for translational inhibition by high adhesion, and that in its absence, an underlying stimulation is evident. This latter effect is consistent with conventional integrin signaling where cell adhesion increases protein synthesis (*Pabla et al., 1999*; *Pola et al., 2013*). To our knowledge, inhibition of protein synthesis at high adhesion has not been reported. However, it is consistent with the correlation between very high adhesion and cellular quiescence in several systems, including vascular smooth muscle cells, senescent cells, and myofibroblasts (*Cho et al., 2004*; *Dimitrijevic-Bussod et al., 1999*; *Drobic et al., 2007*; *Gadbois et al., 1997*; *Koyama et al., 1996*; *Vivar et al., 2016*). Excessive integrin-mediated adhesion can thus exert inhibitory effects on cells.

Lastly, a role for LLPS formation in the inhibition of translation was investigated through use of a Cry2-p130Cas fusion protein, whose irradiation triggers dimerization of the Cry2 moiety to increase valency and promote phase separation. Illumination of cells expressing Cry2-p130Cas triggered the formation of condensates and a decrease in translation, both of which reversed when illumination was terminated.

Together, these results identify a novel LLPS organelle that mediates a functional connection between integrin-mediated adhesion and regulation of protein synthesis. Major questions for future work include- elucidation of the composition and structure of these condensates in greater detail; determining the mechanisms that govern trafficking between the cytoplasm and focal adhesions; elucidating the molecular mechanisms of translational regulation; and determining the role these structures play in biology and medicine.

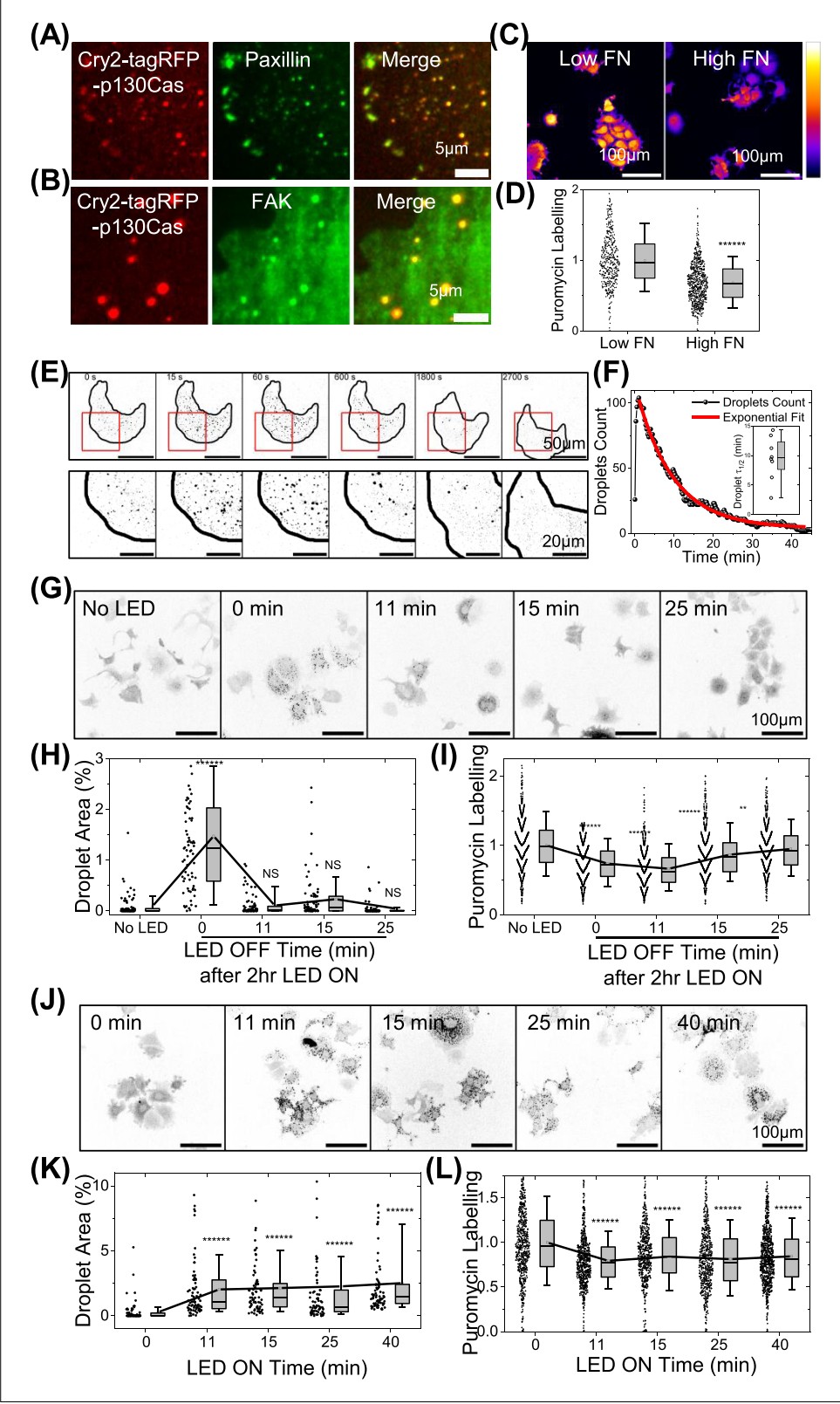

**Figure 8.** Light-induced p130Cas droplets regulate translation. (**A–B**) MCF7 cells expressing light-inducible cry2-tagRFP-p130Cas illuminated with blue LED light (red- left panel) stained for paxillin (**A**) or focal adhesion kinase (FAK) (**B**) (green-middle panel) and merged (right panel). Scale bar = 5 µm. (**C**) Intensity coded images of puromycin labeling in stable MCF7 cell line with modest over-expression of cry2-tagRFP-p130Cas plated on low

*Figure 8 continued on next page*

*Figure 8 continued*

and high fibronectin coated substrate. Scale bar = 100 µm. (**D**) Quantification of puromycin intensity in multiple cells from C. N=446 and 747 cells from 25 field of views for cells on low and high fibronectin, respectively from three independent experiments. (**E**) Time lapse intensity inverted images of MCF7 cells stably expressing cry2-tagRFP-p130Cas before (first frame) and after (second frame onwards) a 0.5 s 488 nm laser pulse. Scale bar = 50 µm. Lower panels show zoomed-in images of the area labeled in the red box. Scale bar = 20 µm. (**F**) Typical plot of the droplets per cell over time after illumination of cry2-tagRFP-p130Cas cells. Red line shows the fit to an exponential function to determine the half-life of the droplets. Inset: Box plot of half-life of p130Cas droplets showing mean at 9.5±3.7 min. N=8 cells. (**G**) Intensity inverted images of cry2-tagRFP-p130Cas cell line with no blue LED light induction (first image) and cells illuminated with pulsed blue LED light for 2 hr then incubated in the dark for indicated times. Cells were then pulsed with puromycin, fixed, and stained. Scale bar = 100 µm. (**H–I**) Quantification of droplet area percentage (**H**) and puromycin labeling intensity (**I**) in cells from (**G**). N>550 cells from 25 FOVS from three independent experiments each for No LED light, 0, 11, 15, and 25 min, respectively after switching off the blue LED after intermittently illumination for 2 hr. (**J**) Intensity inverted images of cry2-tagRFP-p130Cas cell line with no light (first image) or illuminated for with pulsed LED light for the indicated times, labeled with puromycin and stained. Scale bar = 100 µm. (**K–L**) Quantification of droplet area % (**K**) and puromycin labeling intensity (**L**) for cells in (**J**). N>550 cells from 25 FOVS from three independent experiments each for No blue LED light or LED switched on for 11, 15, 25, and 40 min, respectively with intermittent pulses.

The online version of this article includes the following source data and figure supplement(s) for figure 8:

**Figure supplement 1.** Characterization of light inducible p130Cas droplets in cells.

**Figure supplement 1—source data 1.** Original membrane corresponding to *Figure 8—figure supplement 1D*, indicating the relevant bands and treatment.

# Materials and methods
## Cell culture and transfection
Cell lines- NIH3T3 (ATCC), CHO (ATCC), HeLa (ATCC), HEK293T (ATCC), HEK293T cell line stably expressing EGFP-Dcp1A (a gift from Prof. Sarah Slavoff, Department of Chemistry, Yale University) and MCF7 (ATCC) were cultured in Dulbecco's Modified Eagle's Medium (DMEM) (Gibco) with 10% FBS (Gibco) and penicillin-streptomycin (Gibco). MCF10A (ATCC) were cultured in the Mammary Epithelial Basal Medium (MEBM) (Lonza) along with the MEGM SingleQuots Supplements (Lonza) (without GA-1000-gentamycin-amphotericin B mix provided with the kit) and 100 ng/ml Cholera toxin (Sigma Aldrich). Primary HUVECs with each batch composed of cells pooled from three donors, obtained from the Yale Vascular Biology and Therapeutics core facility, were cultured on dishes coated with gelatin (0.1% wt/vol, for 30 min at room temperature (RT) in PBS; Sigma) in M199 (Gibco) with 20% FBS (Gibco), 1 x Penicillin-Streptomycin (Gibco), 60 µg/ml heparin (Sigma: H3393), and endothelial growth cell supplement. ECGS was prepared in Schwartz's laboratory by homogenizing and clarifying the bovine hypothalamus (Pel-Freez Biologicals) as described (*Maciag et al., 1979*). Passage P3-P5 HUVECs were used for experiments.

Cells were cultured in antibiotic-free medium prior to transfection using Lipofectamine 2000 (Invitrogen). EGFP fluorescent tag in p130Cas constructs was the monomeric version of EGFP to reduce artifacts due to tag dimerization. Unless otherwise noted, cells were plated for 6 hr in glass-bottom (MatTek Corporation) dishes coated with 10 µg/ml fibronectin overnight at 4 °C and

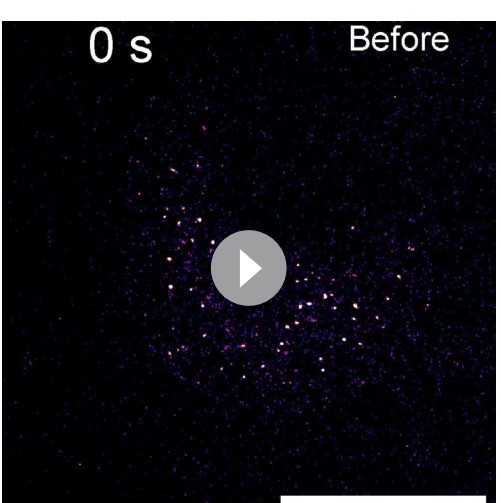

**Video 6.** Movie of induction of p130Cas droplets in MCF7 cell line stably expressing modest levels of cry2-tagRFP-p130Cas upon irradiation with blue laser light for 0.5 s. 0 s frame is before irradiation and 15 s onwards frame is after irradiation. Time points are shown in each image, frame interval = 15 s. Scale bar = 50 µm.

https://elifesciences.org/articles/96157/figures#video6

then imaged. Low, Medium, and High FN refers to fibronectin coating at concentrations 2.5, 10, and 50 μg/ml, respectively.

To make MCF7 cells stably expressing cry2-tagRFP-p130Cas, lentivirus particle-rich medium was collected after 48 hr and centrifuged at 300 g at 4 °C for 15 min to remove particulates. Plasmid-containing wt-cry2 was a gift from Prof. Karla M. Neugebauer, School of Medicine, Yale University. For infection, this supernatant plus 4 μg/ml polybrene was added to MCF7 cells plated overnight at 70% confluency. 18 hr later, infection media were replaced with fresh growth medium. After 2 d, cells were replated and subsequently passaged four more times for stable incorporation of the plasmid while checking the fluorescence levels under the microscope. These cells were FACS sorted (Yale Flow Cytometry Facility, Yale University) based on the signal in the red channel. Furthermore, the expression level was checked by immunoblotting for p130Cas. We periodically tested for mycoplasma and the cell lines negative for mycoplasma were used for the experiments.

## Drug treatments

Cells were treated with 5% (vol %) 1,6-Hexanediol solution (Millipore Sigma), 100 mM Ammonium acetate solution (Millipore Sigma), or 100 μg/ml Cycloheximide (Millipore Sigma) by diluting stock solutions to 2x final concentration in 1 ml culture media maintained at 37 °C and gently adding it to dishes with 1 ml culture media. To assay global mRNA translation, live cells in 1 ml medium received 1 ml medium containing 40 μM puromycin (Thermo Fisher Scientific) at 37 °C, final concentration 20 μM. Cells were incubated at 37 °C for 10 min with intermittent swirling to achieve uniform labeling, followed by immediate fixation with 4% paraformaldehyde in PBS at 37 °C and then immunostaining with anti-puromycin antibody (Sigma Aldrich, clone 12D10, MABE343).

## Immunostaining, antibodies, and RNA FISH

Cells were fixed in 4% for 20 min and permeabilized with 0.1% Triton X-100 in PBS for 20 min at RT. Cells were blocked with 1% BSA in PBS for 1 hr at RT and then incubated with primary antibody at 4 °C overnight. The following primary antibodies were used with 1:500 dilution in 1% BSA in PBS: rabbit polyclonal anti-p130Cas (GeneTex, GTX100605), mouse monoclonal anti-FAK (Sigma-Aldrich, clone 4.47, 05–537), mouse monoclonal anti-Paxillin (BD Biosciences, clone 349, 610051), rabbit polyclonal anti-Phospho Tyr165 p130Cas (GeneTex, GTX132160), rabbit polyclonal anti-G3BP2 (Sigma Aldrich, HPA018425), mouse monoclonal anti-PABP (Santa Cruz Biotechnology, clone 10E10, sc-32318), rabbit monoclonal anti-Argonaute-2 (Abcam, ab156870), rabbit polyclonal anti-DCP1A (Sigma, D5444), and mouse monoclonal anti-Puromycin (Sigma Aldrich, clone 12D10, MABE343). Cells were washed in PBS thrice and then incubated with Alexa Fluor 647–conjugated secondary antibody diluted in PBS (1:1000; Invitrogen) at RT for 1 hr. Nuclei were labeled using Hoechst 33342 (1:1000 in PBS; Molecular Probes). RNA FISH was performed according to the manufacturer's (Stellaris RNA FISH, Biosearch Technologies) protocol for adherent cells using custom Alexa Fluor 488 conjugated at 3' end of Poly-T (30 repeats) probes (IDT-Integrated DNA technologies) (a gift from Prof. Stefania Nicoli, CVRC, Yale University).

## Knockdown and western blot

p130cas knockdown in MCF7 and HUVECs used RNAimax (Invitrogen) transfection of 50 nM human BCAR1 siRNA (9564; ON-TARGETplus, SMARTpool, L-020465-02-0005, Horizon Discovery/ Dharmacon) siRNA; in NIH 3T3 cells, we used mouse BCAR1 siRNA (12927; ON-TARGETplus, SMARTpool, L-041961-00-0005, Horizon Discovery/Dharmacon) or scrambled control siRNA (AM4636; Ambion). Knockdown efficiency and p130Cas over-expression levels were confirmed by Western blotting of cell lysates in radioimmunoprecipitation assay (RIPA) buffer (25 mM Tris-HCl, pH 7.5 [Sigma-Aldrich], 150 mM NaCl [JT Baker], 1% NP-40 [Sigma-Aldrich], 1% sodium deoxycholate [Sigma-Aldrich], and 0.1% SDS [American Bioanalytical] in milliQ water); protease and phosphatase inhibitor (Thermo Fisher Scientific) was added just before extraction. Protein was resolved using SDS-PAGE and transferred to the nitrocellulose membrane using a transfer system (Trans-Blot Turbo; Bio-Rad Laboratories). The membrane was blocked using 5% skimmed milk (American Bioanalytical) in TBS with 0.1% Tween 20 (TBST; Sigma-Aldrich) and incubated in TBST overnight at 4 °C with the following primary antibodies diluted 1:2000: rabbit polyclonal anti-p130Cas (GeneTex, GTX100605), rabbit monoclonal anti-GAPDH (2118, clone 14C10, Cell Signaling Technology), and mouse tubulin (05–829, clone DM1a;

Sigma-Aldrich). Membranes were washed with TBST for 5 min 3x at RT on a shaker. The membrane was then incubated with the appropriate HRP-conjugated secondary antibody (Santa Cruz Biotechnology, Inc) diluted in TBST (1:5000) and visualized using chemiluminescence detection method with Supersignal West Pico (Thermo Fisher Scientific) on the G:Box system (Syngene).

## Imaging and analysis

Imaging and analyses were done essentially as described (*Kumar et al., 2016*). Briefly, imaging was done on an Eclipse Ti microscope (Nikon) equipped with a spinning disk confocal imaging system (Ultraview Vox; PerkinElmer) and an electron-multiplying charged-coupled device camera (C9100-50; Hamamatsu Photonics), using 60×oil objectives for high-resolution images and 20×air objective for low magnification imaging capturing multiple fields of view. During imaging, live cells were maintained at 37 °C with humidity and $CO_2$ control. Images were acquired using Velocity 6.6.1 software. For FRAP, a prebleach image was acquired and then a laser pulse at 100% power of the 488 nm line were used to bleach the entire droplet. Time-lapse images were then acquired every 2 s for 2 min. Images were corrected for photobleaching during image acquisition, and normalized FRAP curves were plotted. Individual recovery curves were in Origin 2018 (64 Bit) (OriginLab Corporation) using single component exponential fit to obtain mobile fraction and timescale of recovery. For imaging droplets, the focal plane through the center of the nucleus, ~2–3 µm above the FAs, was chosen. Planes were combined for representation. The number of droplets were determined by thresholding the image based on intensity and circularity and particles were counted using ImageJ (National Institutes of Health). Student's t-test was performed between the two groups to calculate statistical significance and P-value. At least P-value < 0.05 was considered significant. NS is not significant while *, **, ***, ****, ***** and ****** represent P-value < 0.05, 0.005, 0.0005, 0.00005, 0.000005 and 0.0000005 respectively. For paired Student's t-test , the paired data points are connected by lines (*Figure 1K&M*). Graphs were plotted in Origin 2018–2022 (64 bit).

## Light-induced p130Cas droplets

To estimate p130Cas droplet lifetime, MCF7 cells stably expressing cry2-tagRFP-p130cas were cultured in the dark on glass bottom dishes and imaged on the confocal microscope. After the first frame, a single cell was irradiated using a 488 nm laser (*Courchaine et al., 2021*) for 0.5 s then time-lapse images using 568 nm laser excitation were taken every 15 s for ~45 min. For fixed cell imaging, blue LED (470 nm) (*Che et al., 2015*) light array pulsed induction protocol- schematic shown in *Figure 8—figure supplement 1E* was followed - exposure with 5 s LED ON followed by 60 s OFF time was determined to produce minimal heating and or decreases in cell viability. The LED light array was controlled using a custom-programmable switch (Arduino).

## Isolation of p130Cas droplets and characterization

HEK293T cells were plated overnight in six 150 mm plastic bottom dishes in culture media without antibiotics. Two dishes each were transfected using the manufacturer's protocol with either EGFP-p130cas (25 µg plasmid DNA, 75 µl Lipofectamine 2000 (Invitrogen)) or tagRFP-paxillin (25 µg plasmid DNA, 75 µl Lipofectamine 2000) or control- EGFP (5 µg plasmid DNA, 15 µl Lipofectamine 2000). At 6 hr, media were changed to culture media without antibiotics. After 48 hr, cells were trypsinized and pelleted. Cell pellets were washed twice with 1 x PBS and pellets frozen at –80 °C, then thawed on ice and lysed in ice-cold 200 µl lysis buffer (1% NP-40 (Sigma Aldrich) in DPBS (Thermo Fischer Scientific)), 4 µl protease and phosphatase inhibitor (Thermo Fisher Scientific) and 10 µl RNAseout (Thermo Fisher Scientific) for 30 min on ice with intermittent vortexing. Lysates were spun at 5000 g for 30 min at 4 °C to remove cellular debris. To further clarify, the the supernatant was transferred to a new Eppendorf tube and spun again at 5000 g for 10 min at 4 °C.

Simultaneously, 80 µl GFP trap Dynabeads (Chromotek) were washed with ice-cold lysis buffer 3 x and collected by spinning at 2000 g for min at 4 °C. Buffer was carefully removed using a pipette to avoid loss of beads. Beads were then blocked using 1 ml lysis buffer with 5 µl RNAseOUT and 50 µg yeast tRNA (Thermo Fisher Scientific) for 1 hr at 4 °C with slow rotation. Beads were divided equally

into two Eppendorf tubes and the blocking buffer was removed. 400 µl clarified cell lysate containing EGFP-p130cas or EGFP were added to each tube containing beads along with 200 µl clarified cell lysate containing tagRFP-paxillin and slowly rotated for 2 hr at 4 °C.

100 µl bead suspension was transferred to a new Eppendorf tube for imaging, another 100 µl washed 3x with 1 ml lysis buffer for Western blotting, and the remaining 400 µl fixed using 5 mM reversible cross-linker DSP (dithiobis[succinimidylpropionate]) (Thermo Fisher Scientific) for 30 min at 4 °C with slow rotation. After fixation, beads were washed four times using 1 ml lysis buffer with 1 µl RNAseout and 2 µl protease and phosphatase inhibitor. 200 µl bead samples were eluted in 1x Laemlli buffer (Bio-Rad) with β-Mercaptoethanol (Sigma Aldrich) and boiled at 95 °C for 5 min. The samples were loaded on a 4–20% SDS Tris-Glycine gradient gels (Bio-Rad). When samples entered the gel by ~1 cm, the run was halted. The entered sample was excised out of the whole gel, put in an Eppendorf, and fixed using 45% methanol (Sigma Aldrich) and 10% Acetic Acid (Sigma Aldrich) in water. The gel plug was washed 3 x with ultra clean water (Sigma) before submitting to the Keck MS & Proteomics Resource at Yale School of Medicine for LC MS/MS mass spectrometric analysis. Peptides with at least two counts and twofold enrichment relative to control beads were considered specific and was used for further gene ontology analysis.

To the other 200 µl bead sample, 700 µl Qiazol along with 7 µl β-Mercaptoethanol (Sigma Aldrich) was added and kept at –80 °C. Sample was thawed on ice and RNA was isolated using miRNeasy Kits (Qiagen) according to the manufacturer's protocol and the sample was submitted to Yale Center for Genome Analysis (YCGA) for RNA sequencing.

100 µl bead sample was divided equally into four Eppendorf tubes. One sample was left untreated one treated with 100 mM ammonium acetate solution, one with 5% 1,6-Hexanediol, and one washed 3–4x with ice-cold lysis buffer. 1–2 µl of these samples were imaged under the microscope. Then, all these samples were fixed with 4% PFA, washed 3–4x with ice-cold PBS, and imaged.

## RNA sequence analysis

Bulk RNA-seq analysis was carried out as described (*Kosyakova et al., 2020*). Briefly, for each read, the first 6 and the last nucleotides were trimmed with a Phred score of <20. If, after trimming, the read was shorter than 45 bp, the whole read was discarded. Trimmed reads were mapped to the human reference genome (hg38) with HISAT2 v2.1.0 (*Kim et al., 2015*). Alignments with quality scores below 20 were excluded from further analysis. Gene counts were produced with StringTie v1.3.3b (*Pertea et al., 2015*). Differential expression analysis was conducted and normalized counts were produced using DESeq2 (*Love et al., 2014*). P-values were adjusted for multiple testing using the Benjamini–Hochberg procedure (*Benjamini et al., 2001*). We used miRDeep2 (*Friedländer et al., 2012*) for the miRNA/smRNA-seq analysis. RNAs that were enriched by > twofold with p-values <0.05 were considered positives. Gene ontology analysis was carried out using the online web tool - Database for Annotation, Visualization, and Integrated Discovery (DAVID). Number of proteins versus the corresponding GO term is plotted with fold enrichment depicted by the size of the circle and -log10(P-value) by the color of the circle.

To determine the likelihood of mRNAs present in the p130cas droplet being differentially regulated in cells having the droplet, 46 mRNAs were randomly chosen from the pool of differentially regulated genes in HEK293T cells over-expressing mEGFP-p130Cas using the rand function in Microsoft Excel. This was iterated 10 times and between 0–3 genes of this randomly chosen genes overlapped with the 46 mRNAs determined to be present in the p130Cas droplet (<5%).

## Acknowledgements

We thank Prof. Steven K Hanks (Department of Cell and Developmental Biology, Vanderbilt University, Nashville, Tennessee) and Prof. Sarah Slavoff (Department of Chemistry, Yale University) for providing p130Cas Y15F plasmid and HEK293T cell line stably expressing EGFP-Dcp1A, respectively. We also would like to thank WM Keck Biotechnology Resource Laboratory, Yale Flow Cytometry Facility and Yale Center for Genome Analysis (YCGA) at Yale School of Medicine for LC MS/MS mass spectrometric analysis, cell sorting, and RNA sequencing, respectively. We thank Ho-Joon Lee and Francesc Lopez (Department of Genetics and Yale Center for Genome Analysis, Yale School of Medicine) for help with RNA seq analysis. We thank Rolando Garcia-Milian (Bioinformatics Support Hub, Harvey

Cushing/John Whitney Medical Library, Yale University) for helpful discussion on RNA seq analysis. This work was supported by NIH grant R01 GM047214 and R01 HL135582 to MAS.

## Additional information

### Funding

| Funder | Grant reference number | Author |
|---|---|---|
| National Institutes of Health | R01 GM047214 | Martin A Schwartz |
| National Institutes of Health | R01 HL135582 | Martin A Schwartz |

The funders had no role in study design, data collection and interpretation, or the decision to submit the work for publication.

### Author contributions

Abhishek Kumar, Conceptualization, Resources, Data curation, Software, Formal analysis, Supervision, Validation, Investigation, Visualization, Methodology, Writing – original draft, Project administration, Writing – review and editing; Keiichiro Tanaka, Resources; Martin A Schwartz, Conceptualization, Supervision, Funding acquisition, Investigation, Writing – original draft, Project administration, Writing – review and editing

### Author ORCIDs

Abhishek Kumar ⬚ https://orcid.org/0000-0003-3235-1916
Keiichiro Tanaka ⬚ http://orcid.org/0000-0001-6328-1485
Martin A Schwartz ⬚ http://orcid.org/0000-0002-2071-1243

Reviewer #1 (Public Review): https://doi.org/10.7554/eLife.96157.2.sa1
Reviewer #2 (Public Review): https://doi.org/10.7554/eLife.96157.2.sa2
Author response https://doi.org/10.7554/eLife.96157.2.sa3

## Additional files

### Supplementary files

Supplementary file 1. Excel Sheet containing the list of proteins found in p130Cas droplets determined by mass spectrometry.

Supplementary file 2. Excel Sheet containing the list of mRNAs found in p130Cas droplets determined by RNA sequencing.

Supplementary file 3. Excel Sheet containing the list of differentially regulated mRNAs in HEK293T cells with p130Cas droplets induced by its overexpression as determined by RNA sequencing.

MDAR checklist

### Data availability

All data generated or analysed during this study are included in the manuscript and supporting files; source data files have been provided for Figure 1- figure supplement 1, Figure 7, Figure 7- figure supplement 1 and Figure 8-figure supplement 1.

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
