## [Editor Report · eLife assessment]

In this **valuable** study, Kumar et al., provide evidence suggesting that the p130Cas drives the formation of condensates that sprout from focal adhesions to cytoplasm and suppress translation. Pending further substantiation, this study was found to be likely to provide previously unappreciated insights into the mechanisms linking focal adhesions to the regulation of protein synthesis and was thus considered to be of broad general interest. However, the evidence supporting the proposed model was **incomplete**; additional evidence is warranted to substantiate the relationship between p130Cas condensates and mRNA translation and establish corresponding functional consequences.

---

## [Referee Report · Reviewer #1 (Public Review)]

Summary:

The authors demonstrated the phenomenon of p130Cas, a protein primarily localized at focal adhesions, and its formation of condensates. They identified the constituents within the condensates, which include other focal adhesion proteins, paxillin, and RNAs. Furthermore, they proposed a link between p130Cas condensates and translation.

Strengths:

Adhesion components undergo rapid exchange with the cytoplasm for some unclear biological functions. Given that p130Cas is recognized as a prominent mechanical focal adhesion component, investigating its role in condensate formation, particularly its impact on the translation process, is intriguing and significant.

Weaknesses:

The authors identified the disordered region of p130Cas and investigated the formation of p130Cas condensate. They attempted to demonstrate that p130Cas condensates inhibit translation, but the results did not fully support this assertion. There are several comments below:

(1) Despite isolating p130Cas-GFP protein using GFP-trap beads, the authors cannot conclusively eliminate the possibility of isolating p130Cas from focal adhesions. While the characterization of the GFP-tagged pulls can reveal the proteins and RNAs associated with p130Cas, they need to clarify their intramolecular mechanism of localization within p130Cas droplets. Whether the protein condensates retain their liquid phase or these GFP-p130Cas pulls represent protein aggregate remains uncertain.

(2) The authors utilized hexanediol and ammonium acetate to highlight the phenomenon of p130Cas condensates. Although hexanediol is an inhibitor for hydrophobic interactions and ammonium acetate is a salt, a more thorough explanation of the intramolecular mechanisms underlying p130Cas protein-protein interaction is required. Additionally, given that the size of p130Cas condensates can exceed >100um2, classification is needed to differentiate between p130Cas condensates and protein aggregation.

(3) The connection between p130Cas condensates and translation inhibition appears tenuous. The data only suggests a correlation between p130Cas expression and translation inhibition. Further evidence is required to bolster this hypothesis.

---

## [Referee Report · Reviewer #2 (Public Review)]

Summary:

In this article, Kumar et al., report on a previously unappreciated mechanism of translational regulation whereby p130Cas induces LLPS condensates that then traffic out from focal adhesion into the cytoplasm to modulate mRNA translation. Specifically, the authors employed EGFP-tagged p130Cas constructs, endogenous p130Cas, and p130Cas knockouts and mutants in cell-based systems. These experiments in conjunction with various imaging techniques revealed that p130Cas drives assembly of LLPS condensates in a manner that is largely independent of tyrosine phosphorylation. This was followed by in vitro EGFP-tagged p130Cas-dependent induction of LLPS condensates and determination of their composition by mass spectrometry, which revealed enrichment of proteins involved in RNA metabolism in the condensates. The authors excluded the plausibility that p130Cas-containing condensates co-localize with stress granules or p-bodies. Next, the authors determined mRNA compendium of p130Cas-containing condensates which revealed that they are enriched in transcripts encoding proteins implicated in cell cycle progression, survival, and cell-cell communication. These findings were followed by the authors demonstrating that p130Cas-containing condensates may be implicated in the suppression of protein synthesis using puromycylation assay. Altogether, it was found that this study significantly advances the knowledge pertinent to the understanding of molecular underpinnings of the role of p130Cas and more broadly focal adhesions on cellular function, and to this end, it is likely that this report will be of interest to a broad range of scientists from a wide spectrum of biomedical disciplines including cell, molecular, developmental and cancer biologists.

Strengths:

Altogether, this study was found to be of potentially broad interest inasmuch as it delineates a hitherto unappreciated link between p130Cas, LLPS, and regulation of mRNA translation. More broadly, this report provides unique molecular insights into the previously unappreciated mechanisms of the role of focal adhesions in regulating protein synthesis. Overall, it was thought that the provided data sufficiently supported most of the authors' conclusions. It was also thought that this study incorporates an appropriate balance of imaging, cell and molecular biology, and biochemical techniques, whereby the methodology was found to be largely appropriate.

Weaknesses:

Two major weaknesses of the study were noted. The first issue is related to the experiments establishing the role of p130Cas-driven condensates in translational suppression, whereby it remained unclear whether these effects are affecting global mRNA translation or are specific to the mRNAs contained in the condensates. Moreover, some of the results in this section (e.g., experiments using cycloheximide) may be open to alternative interpretation. The second issue is the apparent lack of functional studies, and although the authors speculate that the described mechanism is likely to mediate the effects of focal adhesions on e.g., quiescence, experimental testing of this tenet was lacking.

---

## [Author Response]

**eLife assessment**
In this valuable study, Kumar et al., provide evidence suggesting that the p130Cas drives the formation of condensates that sprout from focal adhesions to cytoplasm and suppress translation. Pending further substantiation, this study was found to be likely to provide previously unappreciated insights into the mechanisms linking focal adhesions to the regulation of protein synthesis and was thus considered to be of broad general interest. However, the evidence supporting the proposed model was incomplete; additional evidence is warranted to substantiate the relationship between p130Cas condensates and mRNA translation and establish corresponding functional consequences.

We thank the Elife editorial team for their positive assessment of the broad significance of our manuscript. We fully agree that the functional consequences need to be explored in more detail. We feel that many of the criticisms are valid points that are not easily addressed via available tools, thus, should be considered limitations of present approaches. We hope that readers appreciate that identification of a new class of liquid-liquid phase separations calls for much more work to fully explore their characteristics, regulation and function, which will likely advance many areas of cell biology and perhaps even medicine.

**Reviewer #1 (Public Review):**
Summary:The authors demonstrated the phenomenon of p130Cas, a protein primarily localized at focal adhesions, and its formation of condensates. They identified the constituents within the condensates, which include other focal adhesion proteins, paxillin, and RNAs. Furthermore, they proposed a link between p130Cas condensates and translation.Strengths:Adhesion components undergo rapid exchange with the cytoplasm for some unclear biological functions. Given that p130Cas is recognized as a prominent mechanical focal adhesion component, investigating its role in condensate formation, particularly its impact on the translation process, is intriguing and significant.

We thank the reviewer for recognizing the functional significance of the work.

Weaknesses:The authors identified the disordered region of p130Cas and investigated the formation of p130Cas condensate. They attempted to demonstrate that p130Cas condensates inhibit translation, but the results did not fully support this assertion. There are several comments below:(1) Despite isolating p130Cas-GFP protein using GFP-trap beads, the authors cannot conclusively eliminate the possibility of isolating p130Cas from focal adhesions. While the characterization of the GFP-tagged pulls can reveal the proteins and RNAs associated with p130Cas, they need to clarify their intramolecular mechanism of localization within p130Cas droplets. Whether the protein condensates retain their liquid phase or these GFP-p130Cas pulls represent protein aggregate remains uncertain.

We agree, the isolation from cell lysates does not distinguish between focal adhesions and cytoplasmic LLPS. We note that p130Cas in focal adhesions also appears to be in LLPS. But there are no methods available to isolate them separately. We acknowledge this is a limitation of the study.

(2) The authors utilized hexanediol and ammonium acetate to highlight the phenomenon of p130Cas condensates. Although hexanediol is an inhibitor for hydrophobic interactions and ammonium acetate is a salt, a more thorough explanation of the intramolecular mechanisms underlying p130Cas protein-protein interaction is required. Additionally, given that the size of p130Cas condensates can exceed >100um2, classification is needed to differentiate between p130Cas condensates and protein aggregation.

Ammonium acetate, which works by promoting hydrophobic interactions and weak Van der Waals forces, has been widely used in phase separation studies to change ionic strength without altering intracellular pH. Conversely, hexanediol weakens hydrophobic/ Van der Walls interactions that commonly mediate phase separation of IDRs. In the case of p130Cas, the multiple tyrosines and within the scaffolding domain are obvious targets. If the reviewer is asking us to resolve the detailed hydrophobic interactions within the scaffolding domain, this is far beyond the scope of the current paper.

Protein aggregates are defined by their characteristics (e.g irreversibility, departure from spherical) not by size. Older, larger droplets remain circular and show slower but still measurable rates of exchange. Moreover, droplets are essentially absent after trypsinizing and replating cells. All these results argue against aggregates.

(3) The connection between p130Cas condensates and translation inhibition appears tenuous. The data only suggests a correlation between p130Cas expression and translation inhibition. Further evidence is required to bolster this hypothesis.

The optogenetic experiment shows that triggering LLPS by dimerizing p130Cas results in inhibition of translation. This is a causal not a correlative experiment. The reviewer may be thinking that dimerizing p130Cas could stimulate focal adhesion signaling, activating FAK or a src family kinase or other signals. However, none of these signals has been linked to inhibition of cell growth or migration. Thus, we agree that this is a limitation but consider it a low probability mechanism.

**Reviewer #2 (Public Review):**
Summary:In this article, Kumar et al., report on a previously unappreciated mechanism of translational regulation whereby p130Cas induces LLPS condensates that then traffic out from focal adhesion into the cytoplasm to modulate mRNA translation. Specifically, the authors employed EGFP-tagged p130Cas constructs, endogenous p130Cas, and p130Cas knockouts and mutants in cell-based systems. These experiments in conjunction with various imaging techniques revealed that p130Cas drives assembly of LLPS condensates in a manner that is largely independent of tyrosine phosphorylation. This was followed by in vitro EGFP-tagged p130Cas-dependent induction of LLPS condensates and determination of their composition by mass spectrometry, which revealed enrichment of proteins involved in RNA metabolism in the condensates. The authors excluded the plausibility that p130Cas-containing condensates co-localize with stress granules or p-bodies. Next, the authors determined mRNA compendium of p130Cas-containing condensates which revealed that they are enriched in transcripts encoding proteins implicated in cell cycle progression, survival, and cell-cell communication. These findings were followed by the authors demonstrating that p130Cas-containing condensates may be implicated in the suppression of protein synthesis using puromycylation assay. Altogether, it was found that this study significantly advances the knowledge pertinent to the understanding of molecular underpinnings of the role of p130Cas and more broadly focal adhesions on cellular function, and to this end, it is likely that this report will be of interest to a broad range of scientists from a wide spectrum of biomedical disciplines including cell, molecular, developmental and cancer biologists.Strengths:Altogether, this study was found to be of potentially broad interest inasmuch as it delineates a hitherto unappreciated link between p130Cas, LLPS, and regulation of mRNA translation. More broadly, this report provides unique molecular insights into the previously unappreciated mechanisms of the role of focal adhesions in regulating protein synthesis. Overall, it was thought that the provided data sufficiently supported most of the authors' conclusions. It was also thought that this study incorporates an appropriate balance of imaging, cell and molecular biology, and biochemical techniques, whereby the methodology was found to be largely appropriate.

We thank reviewer for this positive assessment.

Weaknesses:Two major weaknesses of the study were noted. The first issue is related to the experiments establishing the role of p130Cas-driven condensates in translational suppression, whereby it remained unclear whether these effects are affecting global mRNA translation or are specific to the mRNAs contained in the condensates. Moreover, some of the results in this section (e.g., experiments using cycloheximide) may be open to alternative interpretation. The second issue is the apparent lack of functional studies, and although the authors speculate that the described mechanism is likely to mediate the effects of focal adhesions on e.g., quiescence, experimental testing of this tenet was lacking.

We appreciate the reviewer’s insights. Assessing translational inhibition for specific genes rather than global measurement of translation is an important direction for future work.

Regarding the cycloheximide experiments, we are unsure what the reviewer means. We used it as a control for puromycin labeling but this is a very standard approach. It seems more likely that the question concerns Fig 5G, where we used it to sequester mRNAs on ribosomes to deplete from other pools. In this case, p130cas condensates decrease after 2 minutes. The reviewer may be suggesting that this effect could be due to blocked translation per se and loss of short-lived proteins. We acknowledge that this is possible but given the very rapid effect (2 min), we think it unlikely.

Lastly, we agree with the reviewer that further functional studies in quiescence or senescence are warranted; however, these are extensive, open-ended studies and we will not be able to include them as part of the current paper.